# Report: The effects of topical pleurotus tuberregium (PT) aqueous extract on intraocular pressure in monkeys

**Ghalib A. Akinlabi** [1] *, **Paul L. Kaufman** [2,3], **Julie A. Kiland** [2]

1 Department of Optometry, University of Benin, Benin City, Nigeria, 2 Department of Ophthalmology and Visual Sciences, University of Wisconsin, Madison, Wisconsin, United States of America, 3 Wisconsin National Primate Research Center, UW, Madison, Wisconsin, United States of America

* gaakinlabi@icloud.com

## Abstract

### Purpose

In earlier experiments in Nigeria, aqueous extract of *Pleurotus tuber-regium* (PT) had been shown to lower intra ocular pressure (IOP) in a feline model. The aim of the current study was to determine whether PT had the same or a similar IOP-lowering effect in ocularly normal non-human primates.

### Methods

Four monkeys were treated twice daily for 4 days with 2 x 20 µl drops of 50 mg/ml PT (pH = 4.3). The monkeys were sedated with 5–10 mg/kg ketamine HCl IM. PT was administered to the right eye and BSS to the left eye. Baseline IOP was measured just prior to beginning treatment, and on day 5 before treatment and then hourly for 3 hours, beginning 1 hour after treatment. SLEs were performed at baseline and on day 5 pre- and 3 hours post-treatment.

### Results

There was no significant difference between IOP in treated vs control eyes in the protocol. There were no adverse effects or toxicity as seen by SLE.

### Conclusions

The inability of the extract to lower IOP in monkeys, in contrast to ocular hypertensive cats in an earlier study, could be due to species differences or duration of treatment. Since no adverse effects were observed in the monkeys, further studies with varying durations and dosages are recommended.

**Data Availability Statement:** All relevant data are within the paper.

**Funding:** This study was sponsored by grants from the Nigerian tertiary education trust fund (TETFUND), Research to Prevent Blindness, New

York (unrestricted departmental grant), NIH/NEI
(Core Grant for Vision Research (P30 EY016665)),
and the Ocular Physiology Research & Education
Foundation. GA: Nigerian tertiary education trust
fund (TETFUND). PK & JK: Research to Prevent
Blindness, New York (unrestricted departmental
grant), NIH/NEI (Core Grant for Vision Research
(P30 EY016665)), and the Ocular Physiology
Research & Education Foundation.

**Competing interests:** The authors have declared
that no competing interests exist.

## Introduction

*Pleurotus tuberregium* (PT) is an edible, gilled mushroom, native to the tropics in Africa, Asia,
and Australasia [1]. In Africa, PT is essential economically and ethno-medicinally, and
although it is found in the wild, PT can be cultivated using lignocellulosic organic wastes to
generate its characteristic sclerotium or fruit bodies that are used as food and medicine by
indigenous people [2,3]. Scientific evidence indicates that PT is useful in treating high blood
pressure, diabetic hypertriglyceredemia, tumours, as well as fungal and bacterial infections in
animals and humans [4–15], but to our knowledge no randomized masked placebo-controlled
trials have been reported. The active compounds in PT may be peptides, polysaccharides, gly-
coproteins, or phytochemicals [15]. Phytochemical analysis shows that PT contains alkaloids,
saponins, flavonoids, tannins, anthraquinnone and phytates [16,17]. D-Mannitol, one of the
main phytochemicals in the pleurotus species, inhibits angiotensin 1 converting enzyme, lead-
ing to an antihypertensive effect [8]. Drugs that possess antihypertensive, antioxidant and cho-
linergic properties may also have the potential to reduce intraocular pressure (IOP) [18,19].

An initial study from Nigeria reported that topical ocular instillation of aqueous mushroom
extract significantly reduced IOP in cats with dexamethasone-induced ocular hypertension
(DIOH) [20]. The effect was significantly greater in cats after treatment with mushroom
extract compared to both cats treated with timolol and to control cats [20]. In a follow-up
study, IOP after treatment with different concentrations of PT extract (2mg, 4mg and 10mg/
ml) was compared to IOP after treatment with latanoprost in the DIOH feline model. The
results showed that the IOP reduction with latanoprost was greater than with any of the differ-
ent extract concentrations [21]. The current study was designed to determine whether the PT
mushroom extract had the same or a similar IOP-lowering effect in ocularly normal non-
human primates.

## Materials and methods

### Preparation and instillation of PT

The PT used in the current study came from three different sources: the Department of
Optometry, Botany and Pharmacology, University of Benin, Benin City, Nigeria; Prof.
Omo-isi's laboratory at North Carolina A & T State University, Greensboro, NC, USA and
Dr. Paul Kaufman's laboratory in the Department of Ophthalmology and Visual Sciences,
University of Wisconsin, Madison, WI, USA. Mushroom crops were produced in shade
houses, harvested by hand and sundried. All water used in cleaning and processing were
purified with an ultra-filtration system. This super fine membrane technology filters partic-
ulate down to 0.025 microns. Powdered PT (27.217 g) from Nigeria was used to prepare the
PT suspension in Dr. Paul Kaufman's lab. The powder was combined with 320 ml distilled
water and the mixture was stirred for 3 days, at room temperature, using a Nuovo stirrer.
The mixture was then filtered using a Nalgene MF75 series polystyrene Lab Filter unit, with
a pore size of 0.45 µm. The filtered extract (150 ml) was heated and stirred until the liquid
evaporated, yielding 3.03 g of a brown solid mass. The final yield was diluted using distilled
water to either 10 mg/ml or 50 mg/ml and had a pH of 4.3. The diluted extract was sterilized
using an autoclave [19,20,21].

### Animals and anesthesia

Research with non-human primates represents a small but indispensable component of bio-
medical research. The scientists in this study are aware of and committed to ensuring the best
possible science with the least possible harm to the animals [22].

The animals were pair- or group-housed in facilities of the Wisconsin National Primate Research Centre (WNPRC). The facility provides the animals with an enriched environment (incl. a multitude of toys and wooden structures. They were kept in standard AALAC- and USDA-approved WNPRC housing and husbandry conditions, under a 12/12hour light/dark cycle. The Animal care staff fed them in their cages twice a day at 7am and 3pm with monkey chow (Mazuri®Monkey Crunch 20Biscuit). The animals' psychological and veterinary welfare is monitored daily by the WNPRC veterinarians, the animal facility staff and the lab's scientists [23,24].

Five cynomolgus monkeys (Macaca fascicularis) were studied. Monkeys were sedated with 5–10 mg/kg ketamine HCl IM (1–10 mg/kg supplemental doses as needed) for treatments, slit-lamp examinations (SLE) and IOP measurements. All experiments were conducted in accordance with the ARVO Statement on the Use of Animals in Research. The protocol was reviewed and approved by a UW-Madison IACUC. All experiments were conducted within the WNPRC, where the animals and the laboratory were housed, literally down the hall from each other. All animals were returned to their colony after the experiment. WNPRC is one of eight NIH-funded regional/national primate research centers in the US. These world-class facilities adhere to the highest standards of animal husbandry, and veterinary care and over-sight, and training/certification of personnel; indeed, they are the global paradigm.

## Protocol 1

The toxicity and potency of PT was examined in 1 monkey. Baseline IOP was measured just prior to beginning treatment, using a slit-lamp mounted Goldmann applanation tonometer and a TONOVET® rebound tonometer (ICare Finland Oy, Vantaa, Finland). The monkey was then treated twice daily for 4 days with 1 x 4 μl drops of 10mg/ml PT (pH = 4.3) adminis-tered topically to the right eye; 1x4μl drops of balanced salt solution (BSS) was administered to the left eye. Each day, IOP measurements started at 9 am after treatment with either PT or BSS, and then hourly for 3 hours. SLEs were done at immediately pre- and 3 hours post- treat-ment. The monkeys were kept in standard AALAC- and USDA-approved WNPRC housing and husbandry conditions, under a 12/12hour light/dark cycle.

## Protocol 2

The above protocol was repeated in another monkey (n = 1), however the concentration of PT was increased to 50mg/ml and the pH adjusted to 7.2 using NaOH. The monkey was treated twice daily for 4 days with 2 x 20μl drops of 50 mg/ml PT to one eye; 2 x 20 μl drops of BSS were administered to the opposite eye (at 1- minute interval between the drops). Baseline IOP was measured, as above, just prior to beginning treatment. On day 5, IOP was measured just prior to treatment with either PT or BSS and then hourly for 3 hours post-treatment. SLEs were performed immediately pre- and at 3 hours post-treatment.

## Protocol 3

Four cynomolgus monkeys were studied. Baseline SLE, pupil diameter and IOP were mea-sured prior to beginning treatment. Monkeys were then treated twice daily for 4 days with 2 x 20 μl drops of 50 mg/ml PT (pH = 4.3) administered topically to one eye; 2 x 20 μl BSS was administered to the opposite eye (at 1- minute interval between the drops). On day 5, IOP was measured just prior to treatment with either PT or BSS and then hourly for 4 hours. SLEs were done immediately pre- and at 4 hours post-treatment. The results were analyzed using Excel statistical package. IOP data are expressed as the mean ± SEM and were analyzed by the two-tailed paired t-test for ratios compared to 1.0 or for differences compared with 0.0.

## Results

### Protocol 1

There was no significant effect on IOP after 4.5 days treatment with 1 x 4 µl PT. Slight hyperemia was noted in the treated eye on the second day of treatment but had resolved by the third day. No abnormalities were noted in either eye on the fifth day of treatment (Fig 1).

### Protocol 2

No adverse effects were observed after five days of treatment with 2x20µl of 50 mg/ml topical PT. IOP was lower in the PT-treated eye compared to ipsilateral baseline at hour 0 on day 5 (14.5 vs. 19 mmHg) and compared to the control eye (14.5 vs. 18mmHg). However, there was no clear difference when comparing the treated eye to ipsilateral baseline or to the control eye at any other time point on day 5 (Fig 2).

### Protocol 3

No adverse effects were observed after five days of treatment with 2x20µl of 50 mg/ml topical PT in any monkeys. Data from the monkey treated in protocol 2 were combined with data from the 4 monkeys treated in protocol 3 since all received the same dose (50 mg/ml PT or

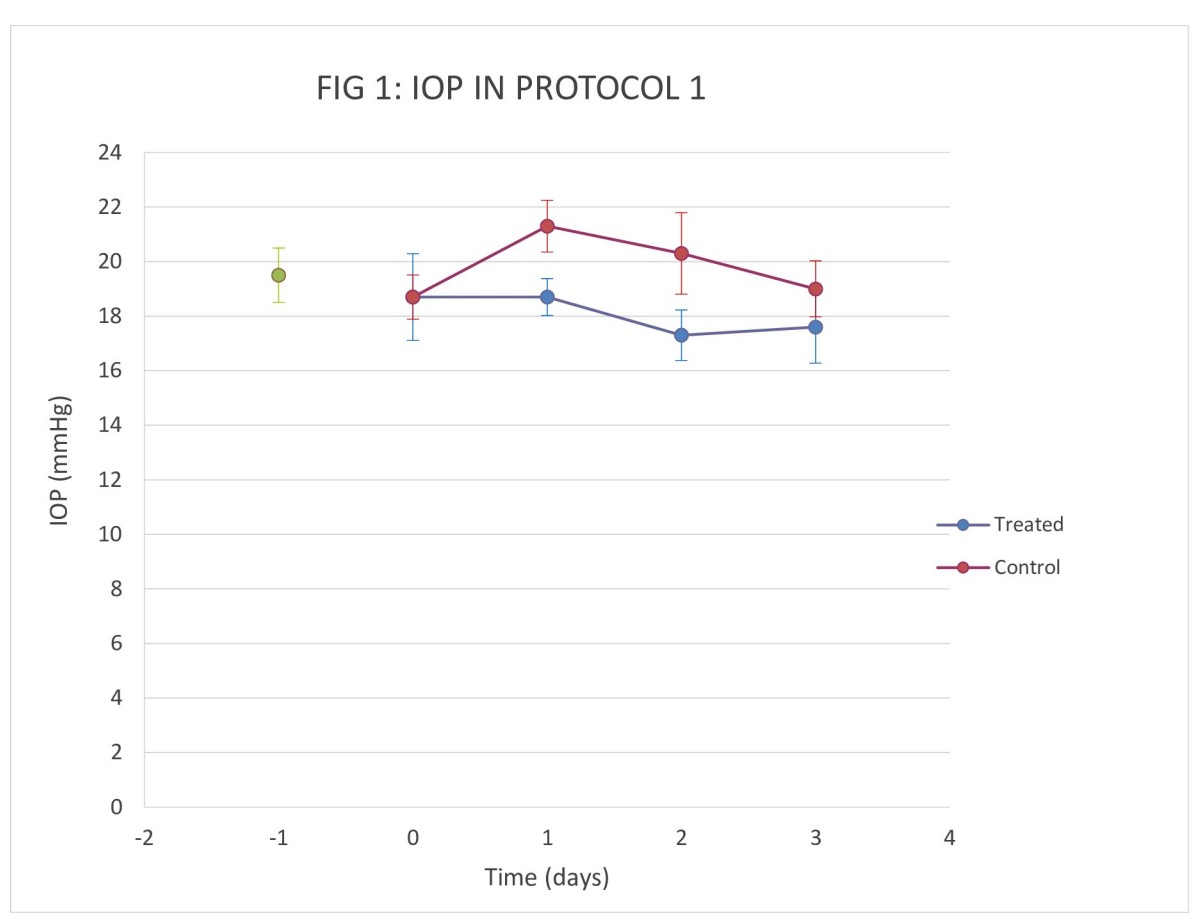

**Fig 1. Comparison of change in IOP in protocol 1 (n = 1).** IOP was measured each day after treatment with 10mg/ml of PT (OD) and BSS as control (OS). Data are mean +/- SD.

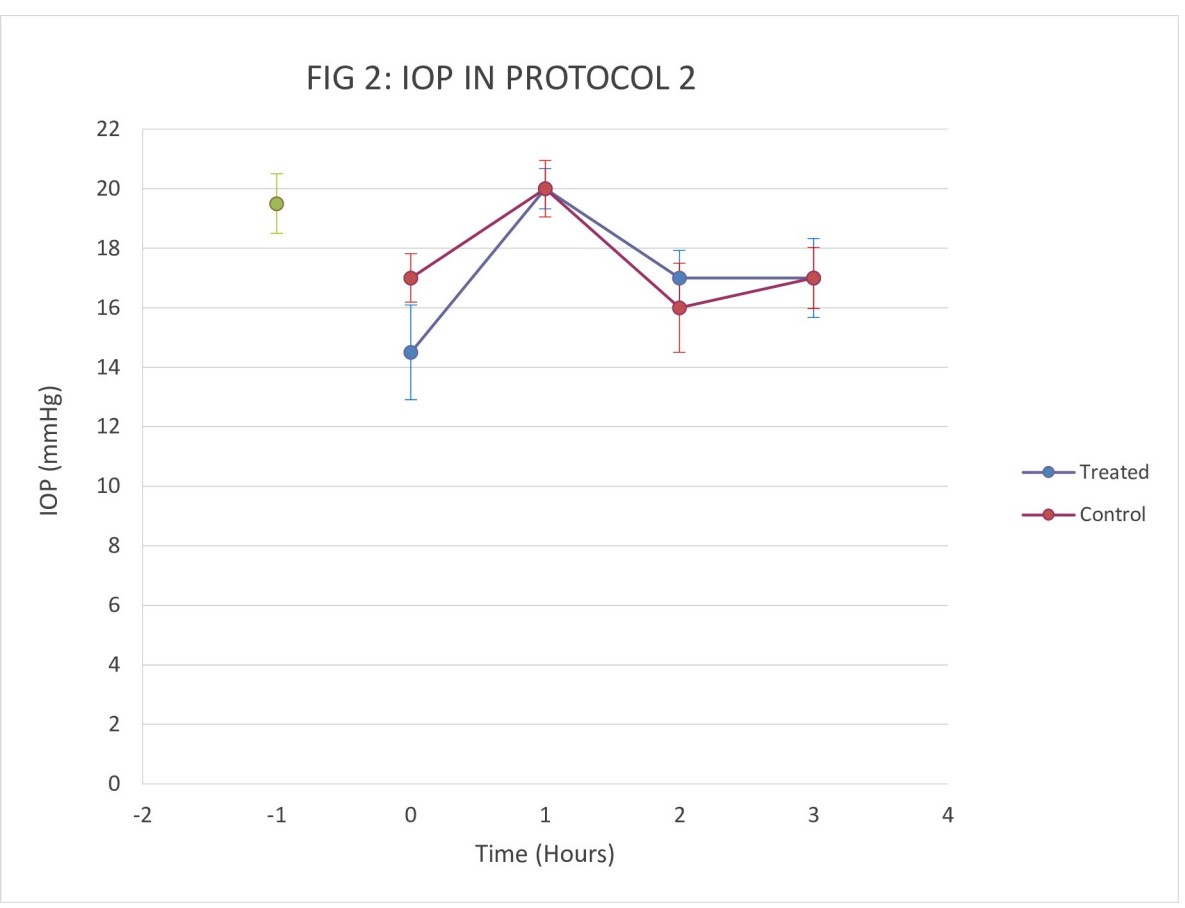

**Fig 2. Comparison of change in IOP in protocol 2 (n = 1), after 4.5 days of treatment with 50mg/ml of PT (OD) and BSS as control (OS).** Data are mean +/- SD.

BSS) topically twice daily over 4.5 days for a total of 9 treatments. There was no significant difference between mean IOP when comparing treated eyes to ipsilateral baseline or to control eyes at any time point (Fig 3).

There was no significant difference between the IOP of the treated and control eyes, when we compared the mean change in IOP of the five cynomolgus monkeys after twice daily administration of 2x20µl of 50mg/ml of PT to one eye, BSS to the opposite eye, for 4.5 days (Fig 4).

Analysis of variance also showed no significant difference between the treated (T) and control (C) IOP values or between treated and baseline (BL) IOP values (Table 1). ANOVA analysis of the PT mushroom extract IOP data over time using Instat GraphPad Prism revealed no significant differences [(T-BL)—(C-BL); p = 0.8464 and treated vs control over time; p = 0.5187]. We also ran the ANOVA for (T-BL)—(C-BL) using the data analysis package in MS Excel and got the same results (p = 0.8464).

## Discussion

Topical administration of PT at 10 and 50 mg/ml caused no obvious adverse effects in the eyes of monkeys as evidenced by slit-lamp examination, other than transient hyperemia on day 2 of treatment in one monkey. This corroborates earlier studies indicating that topical PT is not toxic [25]. Changing the pH of PT from 4.3 to 7.2 did not appear to affect the SLE or IOP in

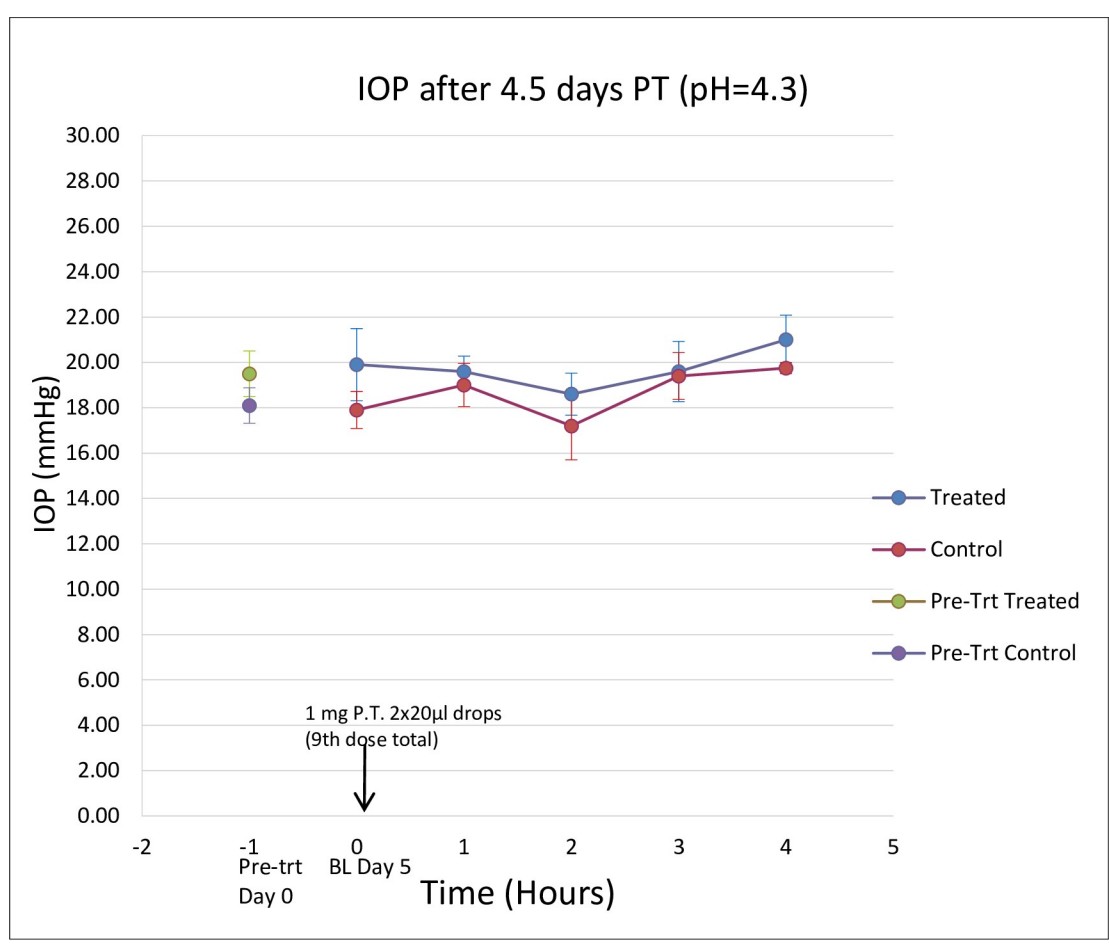

**Fig 3. Comparison of change in IOP in protocol 3 (n = 5), after 4.5 days of treatment with 50mg/ml of PT (OD) and BSS as control (OS).** Data are mean +/- SD (n = 5).

the monkeys. However, evidence of discomfort or a 'stinging' sensation from the eye drops, as evidenced by squinting, increased blinking, or rubbing of the eyes, may have been masked since the animals were sedated.

In previous experiments, PT reduced steroid-induced ocular hypertension in cats [20,21] and also contracted bovine iris muscle *in* vitro [19]. The contractile effect of PT on muscarinic receptor-containing iris muscle may explain its IOP reducing property [19]. However, it must be remembered that the iris has nothing to do with aqueous outflow. In live monkeys, the iris can be completely removed, and IOP, outflow facility, and the facility response to pilocarpine are unchanged [26–28]. If there is muscarinic receptor-mediated iris muscle contraction, there could also be muscarinic receptor-mediated contraction of the ciliary muscle in monkeys, and that might increase facility. This is only one *possible* mechanism; any of the other parameters affecting aqueous humor formation or outflow could be at play. One has to start somewhere. Aside from that we are agnostic as to the mechanism. Assuming that PT lowers IOP in live NHP, we would identify the individual physiological parameter affected (aqueous humor formation, conventional outflow, uveoscleral outflow, Schlemm's canal and episcleral venous pressure, etc. These parameters can each be measured [29–33].

In the current study, PT did not lower IOP in ocular normotensive monkeys. The extract had no effect on the IOP in our small group of normal eyes, but there is a possibility that the

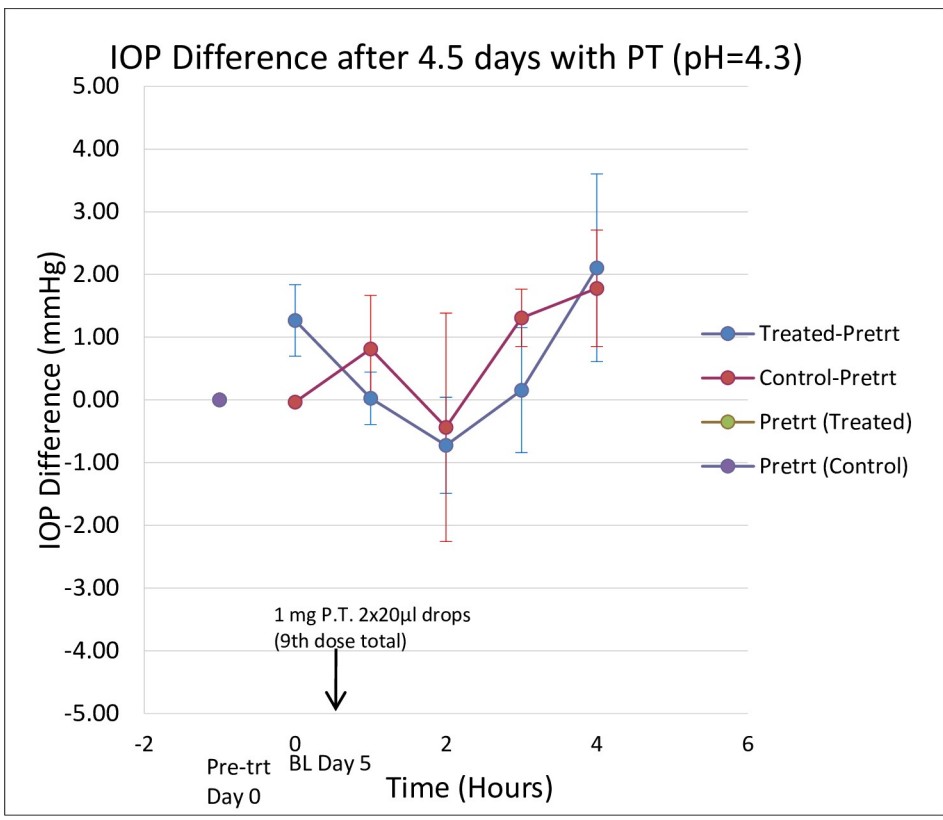

**Fig 4. Comparison of the mean change in IOP in a group of five cynomolgus monkeys' eyes after twice daily administration of 50mg/ml of PT for 4.5 days.** The data showed the mean change in IOP treatment-pretreatment for the treated and control eyes. There is no significant difference between the IOP of the treated and control eyes (Fig 3 and Table 1). Data are mean +/- SEM.

study group was too small to detect a small effect. Sample size calculation for paired organs conducted based on the mean and standard error of the current data indicated that at n equals 5 to 8, we would have sufficient power to detect a physiologic response greater than or equal to 25% of baseline for a two-sided test and 5% significance [25,26]. The calculation was for outflow facility; theoretically, IOP might be different, although IOP is linked to outflow facility. While our data set is small, this preliminary experiment tells us that we are not likely missing a big effect that occurs across the board in all animals. Whether a higher dose or longer duration

**Table 1. Analysis of variance (ANOVA).**

| Test | Ordinary ANOVA (pre-trt through h 4, d 5) | Repeated measures ANOVA (pre-trt through h3, d 5) |
|---|---|---|
| Treated | p = 0.8073 | P = 0.5158 |
| Control | p = 0.4719 | P = 0.4326 |
| Trt-Cont | p = 0.7919 | P = 0.5131 |
| Trt-BL (day 5, hour 0) | p = 0.9933 | P = 0.8221 |
| Cont-BL (day 5, hour 0) | p = 0.2414 | P = 0.3097 |
| (Trt-pretrt)-(Cont-pretrt) [day 0] | p = 0.4366 | P = 0.1068 |
| (T-BL)-(C-BL) [day 5] | P = 0.8464 | P = 0.1284 |

would have mattered we cannot say, but this is not like pilocarpine, epinephrine, rho kinase inhibitors, timolol, nitric oxide donors, $PGF_{2\alpha}$ analogues, etc., where there is a big effect after short-term treatment in most human or monkey subjects [34–43].

There are many possible reasons for this. Depending on the mechanism of action of PT, any small increase in outflow or decrease in aqueous formation may not have been large enough to have a significant effect on IOP in normal monkeys but might have had a greater effect in ocular hypertensive cats [20]. Another possible reason could be anatomical and/or physiological differences between monkey and cat eyes. It is also possible that the extract used in this study was not effective (it was a different batch from that used in cats) or that the previous results in cats were anomalous. Another possible explanation is that the duration of treatment in cats was longer than it was in monkeys. IOP measurements were done in cats after 2 weeks of twice-daily administration of 10mg/ml of PT but were done in monkeys after only 4.5 days of twice-daily treatment. Lastly, there is also the possibility of observer bias in the cat experiments because the observer in that study knew which eyes were treated and which were control. While we realize the very small data set, the fact that there was not even a hint of a signal precluded additional experiments in a scarce precious resource, given that there is unlikely to be a glaucoma therapeutically useful IOP-lowering effect in humans. It is of course possible that there is a physiological effect too small to detect in so few animals.

The next stages in this research will include NMR or mass spectroscopy to check the stability of the extract, additional in vivo experiments with different doses of PT, and increasing the duration of treatment. If an IOP effect is present, we may then investigate the effect of PT on aqueous humor formation and drainage. Earlier work shows that PT extract reduces IOP in dexamethasone-induced glaucoma in a feline model [20,21]. Mechanistic studies would be needed to uncover an effect of PT on aqueous humor production/outflow (trabecular meshwork/Schlemm's canal. EVP, uveoscleral outflow, etc.), and then to analyze morphology/ultrastructure, etc. in primates. This is far, far beyond the scope of the current exploratory study.

We will study the effect of ethanolic extract on IOP. It has been reported that phytochemical analysis of ethanolic and aqueous extract of P. tuber-regium reveals that traces of polyphenols and saponins were more in ethanolic than aqueous extract, although alkaloids, glycosides, saponins, flavonoids, tannins and polyphenol, were present in both aqueous and ethanolic extract [44].

## Author Contributions

**Conceptualization:** Ghalib A. Akinlabi.

**Data curation:** Paul L. Kaufman.

**Formal analysis:** Julie A. Kiland.

**Funding acquisition:** Ghalib A. Akinlabi, Paul L. Kaufman.

**Investigation:** Ghalib A. Akinlabi, Paul L. Kaufman, Julie A. Kiland.

**Methodology:** Ghalib A. Akinlabi, Paul L. Kaufman, Julie A. Kiland.

**Project administration:** Paul L. Kaufman, Julie A. Kiland.

**Resources:** Ghalib A. Akinlabi, Paul L. Kaufman.

**Supervision:** Paul L. Kaufman.

**Validation:** Paul L. Kaufman.

**Visualization:** Ghalib A. Akinlabi.

**Writing – original draft:** Ghalib A. Akinlabi.

**Writing – review & editing:** Paul L. Kaufman, Julie A. Kiland.

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
