## [Decision Letter · Decision Letter 0]

6 Jan 2021

PONE-D-20-33452

Report: The effects of topical Pleurotus tuber-regium (PT) aqueous extract on intraocular pressure in monkeys

PLOS ONE

Dear Dr. AKINLABI,

Thank you for submitting your manuscript to PLOS ONE. After careful consideration, we feel that it has merit but does not fully meet PLOS ONE’s publication criteria as it currently stands. Therefore, we invite you to submit a revised version of the manuscript that addresses the points raised during the review process.

The study on topical use of aqueous extract of Pleurotus tuberregium mushroom on intra-ocular pressure in monkeys is a interesting study. However, there are some concerns in the study which is elaborated in the comments of the reviewers. Moreover, the authors have studied only the effect of Aqueous extracts, which contains only glucans and proteins but alkaloids, saponins, flavonoids, tannins, anthraquinnone, etc can not be extracted in water extracts. Thus the study do not represent the complete effect of PT.

We look forward to receiving your revised manuscript.

Kind regards,

Shwet Kamal, Ph.D

Academic Editor

PLOS ONE

Journal Requirements:

2. In order to comply with PLOS ONE's guidelines for non-human primate experiments (http://journals.plos.org/plosone/s/submission-guidelines#loc-non-human-primates), we kindly request details regarding housing conditions, feeding regimens, environmental enrichment, and all relevant steps taken to alleviate suffering (anesthesia, analgesia, details about humane endpoints, euthanasia, etc.). Please also indicate how often animal care staff monitored the health and well-being of the animals and the criteria used to make such assessments. Lastly, please specify the disposition of animals at the end of the study (e.g. euthanasia, returned to home colony, etc.). If animals were euthanized following the study, please provide the method of sacrifice.

3. Please ensure that you refer to Figure 1 in your text as, if accepted, production will need this reference to link the reader to the figure.

4. Please include your tables as part of your main manuscript and remove the individual files. Please note that supplementary tables (should remain/ be uploaded) as separate "supporting information" files.

Additional Editor Comments (if provided):

The study on topical use of aqueous extract of Pleurotus tuberregium mushroom on intra-ocular pressure in monkeys is a interesting study. However, there are some concerns in the study which is elaborated in the comments of the reviewers. Moreover, the authors have studied only the effect of Aqueous extracts, which contains only glucans and proteins but alkaloids, saponins, flavonoids, tannins, anthraquinnone, etc can not be extracted in water extracts. Thus the study do not represent the complete effect of PT.

The authors are requested to re-submit a revised version in the light of the comments of the reviewers.

Reviewers' comments:

Reviewer's Responses to Questions

**Comments to the Author**

1. Is the manuscript technically sound, and do the data support the conclusions?

Reviewer #1: No

Reviewer #2: Yes

2. Has the statistical analysis been performed appropriately and rigorously? 

Reviewer #1: N/A

Reviewer #2: Yes

3. Have the authors made all data underlying the findings in their manuscript fully available?

Reviewer #1: Yes

Reviewer #2: Yes

4. Is the manuscript presented in an intelligible fashion and written in standard English?

Reviewer #1: Yes

Reviewer #2: Yes

5. Review Comments to the Author

Reviewer #1: Dear Authors, thank you for the opportunity to read about this interesting line of research.

I find this are highly relevant, however there are several major concerns regarding your study and presentation:

1. In the introduction you mention "...Scientific evidence indicates that PT is useful in treating ... ". The references provided after that are only for the in vitro and in vivo animal works. Pleas provide references to placebo-controlled clinical trials to support this statement. Otherwise it is highly misleading.

2. The scientific premise for the work is not clear. There is no suggested mechanism of action for the PT extract in lowering IOP.

3. What standardization and quality control procedures were used to normalize PT preparations and guarantee reproducibility of the studies?

4. For the IOP measurements - what time of day were they performed? Could you also describe the housing of animals and their expected circadian cycle?

5. In the discussion you make the conclusion that that the "extract has no effect on the IOP in normal eyes" and also make the statement that "there is a possibility that the study groups were small to detect any effect." These statements are contradicting - please correct, with the statistical power if needed.

6. What is the expected mechanism of action for the PT extract?

7. There are several factors that contribute to IOP regulation - could you elaborate on the actual effect of PT extract on the aqueos humour production/outflow, TM state, etc. Also, do you expect it to have differnt effect in dexamethasone-induced glaucoma and healthy eyes?

Reviewer #2: Manuscript is good

Few typographical or grammatical errors need to be checked

Such as in abstract Line no 27. Correct the spelling of Pleurotus tuberregium

Similarly line no. 54 lignocellulosic in place of lognocellulosic

Correct these small errors

6. PLOS authors have the option to publish the peer review history of their article (what does this mean?). If published, this will include your full peer review and any attached files.

Reviewer #1: No

Reviewer #2: No

---

## [Author Response · Author response to Decision Letter 0]

19 May 2021

RESPONSE TO REVIEWERS

Thank you for favorably considering our manuscript and for the corrections pointed out therein. 

Below are the answers to the questions raised by the reviewers.

Issues raised by academic editor

The study on topical use of aqueous extract of Pleurotus tuberregium mushroom on intra-ocular pressure in monkeys is a interesting study. However, there are some concerns in the study which is elaborated in the comments of the reviewers. Moreover, the authors have studied only the effect of Aqueous extracts, which contains only glucans and proteins but alkaloids, saponins, flavonoids, tannins, anthraquinnone, etc can not be extracted in water extracts. Thus the study do not represent the complete effect of PT.

Response

We will study the effect of ethanolic extract on IOP. It has been reported that phytochemical analysis of ethanolic and aqueous extract of P. tuber-regium reveals that traces of polyphenols and saponins were more in ethanolic than aqueous extract, although alkaloids, glycosides, saponins, flavonoids, tannins and polyphenol, were present in both aqueous and ethanolic extract41 (line 227 - 230).

Reviewer #1: 

1. In the introduction you mention "...Scientific evidence indicates that PT is useful in treating ... ". The references provided after that are only for the in vitro and in vivo animal works. Please provide references to placebo-controlled clinical trials to support this statement. Otherwise it is highly misleading.

Response

The sentence has been changed to ‘Scientific evidence indicates that PT is useful in treating high blood pressure, diabetic hypertriglyceredemia, tumours, as well as fungal and bacterial infections in animals and humans4-15, but to our knowledge no randomized masked placebo-controlled trials have been reported.’ (line 55-58)

2. The scientific premise for the work is not clear. There is no suggested mechanism of action for the PT extract in lowering IOP.

Response

Mechanistic studies will be carried out later. PT was reported to contract muscarinic receptor-containing bovine iris muscle in an organ bath preparation19. However, it must be remembered that the iris has nothing to do with aqueous outflow. In live monkeys, the iris can be completely removed, and IOP, outflow facility, and the facility response to pilocarpine are unchanged23,24,25. If there is muscarinic receptor-mediated iris muscle contraction, there could also be muscarinic receptor-mediated contraction of the ciliary muscle in monkeys, and that might increase facility. Aside from that we are agnostic as to the mechanism. Assuming that PT lowers IOP in live NHP, we would identify the individual physiological parameter affected (aqueous humor formation, conventional outflow, uveoscleral outflow, Schlemm’s canal and episcleral venous pressure, etc. These parameters can each be measured 26-30. (Line 180-190).

3. What standardization and quality control procedures were used to normalize PT preparations and guarantee reproducibility of the studies?

Response

Mushroom crops were produced in shade houses, harvested by hand and sundried. All water used in cleaning and processing were purified with an ultra-filtration system.This super fine membrane technology filters particulate down to 0.025 microns. (Line 86-88). 

4. For the IOP measurements - what time of day were they performed? Could you also describe the housing of animals and their expected circadian cycle?

Response

IOP measurements started at 9 am after treatment and every hour after that, for 3 hours (line 114). 

The monkeys were kept in standard AALAC- and USDA-approved WNPRC housing and husbandry conditions, under a 12/12hour light/dark cycle. (Line 115-117).

5. In the discussion you make the conclusion that that the "extract has no effect on the IOP in normal eyes" and also make the statement that "there is a possibility that the study groups were small to detect any effect." These statements are contradicting - please correct, with the statistical power if needed.

Response

The wording has been changed to: 

The extract had no effect on the IOP in our small group of normal eyes, but there is a possibility that the study group was too small to detect a small effect. Sample size calculation for paired organs conducted based on the mean and standard error of the current data indicated that at n equals 5 to 8, we would have sufficient power to detect a physiologic response greater than or equal to 25% of baseline for a two-sided test and 5% significance22,23. The calculation was for outflow facility; theoretically, IOP might be different, although IOP is linked to outflow facility. While our data set is small, this preliminary experiment tells us that we are not likely missing a big effect that occurs across the board in all animals. Whether a higher dose or longer duration would have mattered we cannot say, but this is not like pilocarpine, epinephrine, rho kinase inhibitors, timolol, nitric oxide donators, PGF2⍺ analogues, etc., where there is a big effect after short-term treatment in most human or monkey subjects31-40. (Line 180-190).

6. What is the expected mechanism of action for the PT extract?

Response

The contractile effect of PT on muscarinic receptor-containing iris muscle may explain its IOP reducing property19. If there is muscarinic receptor-mediated iris muscle contraction, there could also be muscarinic receptor-mediated contraction of the ciliary muscle in monkeys, and that might increase facility. This is only one possible mechanism; any of the other parameters affecting aqueous humor formation or outflow could be at play. One has to start somewhere……](Line 179 – 186).

7. There are several factors that contribute to IOP regulation - could you elaborate on the actual effect of PT extract on the aqueous humour production/outflow, TM state, etc. Also, do you expect it to have different effect in dexamethasone-induced glaucoma and healthy eyes?

Response

Earlier work shows that PT extract reduces IOP in dexamethasone-induced glaucoma in a feline model20,21. Mechanistic studies would be needed to uncover an effect of PT on aqueous humor production/outflow (trabecular meshwork/ Schlemm’s canal. EVP, uveoscleral outflow, etc.), and then to analyze morphology/ultrastructure, etc. in primates. This is far, far beyond the scope of the current exploratory study. (Line 222 – 226)

Reviewer #2: Manuscript is good

Few typographical or grammatical errors need to be checked

Such as in abstract Line no 27. Correct the spelling of Pleurotus tuberregium

Similarly line no. 54 lignocellulosic in place of lognocellulosic

Correct these small errors

Response

The errors have been corrected (Line 27 and 54).

Yours truly,

G.A. Akinlabi

---

## [Editor Report · Decision Letter 1]

9 Aug 2021

Report: The effects of topical Pleurotus tuber-regium (PT) aqueous extract on intraocular pressure in monkeys

PONE-D-20-33452R1

Dear Dr. AKINLABI,

We’re pleased to inform you that your manuscript has been judged scientifically suitable for publication and will be formally accepted for publication once it meets all outstanding technical requirements.

Kind regards,

Shwet Kamal, Ph.D

Academic Editor

PLOS ONE
---

## [Editor Report · Acceptance letter]

13 Aug 2021

PONE-D-20-33452R1 

Report: The effects of topical pleurotus tuberregium (PT) aqueous extract on intraocular pressure in monkeys 

Dear Dr. Akinlabi:

I'm pleased to inform you that your manuscript has been deemed suitable for publication in PLOS ONE. Congratulations! Your manuscript is now with our production department. 

Kind regards, 

on behalf of

Dr. Shwet Kamal 

Academic Editor

PLOS ONE